# Mental health in Ireland during the Covid pandemic: Evidence from two longitudinal surveys

**David Madden** [ORCID] *

School of Economics, University College Dublin, Dublin, Ireland

* david.madden@ucd.ie

## Abstract

### Background

The Covid pandemic arrived in Ireland on February 29, 2020. In the following weeks various restrictions were introduced to stem the spread of the disease. Anxiety over the spread of the disease and over the restrictions introduced had an adverse effect upon mental health. This study examines the change in mental health for two groups: young adults aged around 23 at the time of onset of Covid (the 1998 cohort) and a sample of principal carers (PCs) of children who were aged 13 at the onset of Covid (the 2008 cohort).

### Methods

Data were obtained from the two cohorts of the longitudinal Growing Up In Ireland (GUI) survey. The sample included 1953 young adults (from the 1998 cohort) and 3547 principal carers (from the 2008 cohort). Mental health as measured by the Centre for Epidemiological Studies Depression—8 scale was obtained for the last pre-Covid wave and for the Covid wave (surveyed in December 2020). Observations for which CES-D8 was not available in either pre or post Covid waves were excluded. Post-Covid sampling weights were applied. The change in depression rates was decomposed into a growth and distribution effect using a Shapley decomposition. The socioeconomic gradient of CES-D8 was examined pre and post Covid using concentration indices and a transition matrix was constructed to examine the dynamics of changes in CES-D8 and depression pre and post-Covid.

### Results

Relative to the last pre-Covid survey, mental health, as measured by CES-D8 deteriorated for both the young adults of the 1998 cohort and the PCs of the 2008 cohort. For young adults, the deterioration was more pronounced for females. There was no observable socio-economic gradient for poor mental health amongst young adults, both pre and post Covid. For mothers from the 2008 cohort, a gradient was observed during the pre-COVID-19 pandemic period with poorer mental health status for lower-income and less educated mothers. This gradient was less pronounced post-Covid, the levelling-off arising from a greater deterioration in mental health for higher-income and better-educated PCs.

**Data Availability Statement:** Data are held in a public repository:The Irish Social Science Data Archive. https://www.ucd.ie/issda/data/guicovid19/
.

**Funding:** The author received no specific funding for this work.

**Competing interests:** The authors have declared that no competing interests exist.

## Conclusion

Both observed cohorts showed a significant deterioration in mental health post Covid. For young adults the effect was significantly more pronounced among females and this is consistent with generally poorer mental health amongst females in this age group. There was little or no socioeconomic gradient observed for young adults, but the gradient became more shallow for principal carers. Care must be taken in terms of drawing policy implications from this study as the Covid-19 pandemic was arguably a unique event, even allowing for the likelihood of future pandemics. However, the study highlights the vulnerability of young adults, especially females, to the mental health effects arising from major public health shocks.

## Introduction

The Covid 19 Pandemic officially arrived in Ireland on February 29, 2020 with the first confirmation of a positive case. Over subsequent weeks various restrictions were introduced to stem the spread of the disease (becoming collectively known as the "lockdown"). These included the closure of all educational establishments and childcare facilities, the banning of various sporting and cultural events and then on March 27, everyone, apart from providers of essential care and services, were advised to stay at home apart from essential visits (e.g. to the supermarket) and exercise within a 2km radius. There was a ban on non-essential travel and on meeting people outside the immediate household.

As Covid cases declined over the summer of 2020 there was a gradual removal of the most severe of these restrictions, but an upsurge in autumn 2020 led to a reimposition of high level restrictions in October. As the second wave of Covid receded there was an easing of restrictions from early December with the opening of non-essential shops and services, and by December 18 limited within-country travel and household visits were permitted. However, there was a significant resurgence of cases in the immediate run-up to and aftermath of Christmas and severe restrictions were again imposed in January 2021.

The various lockdowns were successful in limiting the spread of Covid but these measures were not without their own costs. The most severe restrictions inevitably led to a reduction in economic activity and the reduction in human contact and the hardship imposed by social distancing raises concerns about the possible adverse mental health effects. The latter phenomenon is investigated in this paper using the *Growing Up in Ireland* (GUI) longitudinal survey, which provides mental health information for young adults both before and during the pandemic and also for a sample of Principal Carers (PCs) of a younger cohort (in almost all cases the PCs are the biological mothers).

GUI provided a snapshot of mental health for these groups at a time when restrictions were still relatively tight and had been ongoing for nearly nine months. Critically, since this is a longitudinal study there is similar information *for the same people* for a period *before* Covid.

These groups are by no means representative of the whole nation, though the 2008 and 1998 cohorts are representative of those people born in those specific periods. Nevertheless, what may be lacking in national representativeness is compensated for by having the same measure, for the same observations before and during the pandemic. The change in mental health over this period is investigated, whether this change differed by gender and socioeconomic status and also the pattern of mobility for those people who did experience changes in their mental health.

A number of studies have examined the impact of Covid restriction on various measures of mental health or well-being. In general, negative effects on mental health were found in the early stages of the pandemic, but much of this impact had abated by the middle of the summer of 2020 [1, 2]. In some cases the impact was worst for young adults, particularly women [3].

In the later stages of the pandemic, a number of authors found further negative effects upon mental health for the second and third "waves" of Covid 19 [4–6]. Again, effects were more pronounced for younger people and females, with an effect also found for those with children. However the latter study did not have information on mental health pre-pandemic.

Finally, some studies for Ireland [7] found no evidence of an increase in mental health problems during the first year of the pandemic. However, information on mental health before the pandemic was lacking, and so while mental health may not have deteriorated subsequent to the arrival of Covid 19, it is not possible to tell if the pandemic caused an immediate deterioration in mental health.

[8] used the GUI Covid module to investigate mental health for the 2008 cohort. However, since they did not have a consistent measure of mental health pre and post Covid their study was unable to explicitly examine how mental health changed with the pandemic. They analysed the association between the level of mental health in December 2020 and various factors such as family financial and education resources and restrictions on social activities. They found that lower mental health for the 2008 cohort was associated with a fall in family income arising from the pandemic (as opposed to family income pre-pandemic) and also with lower educational resources in terms of access to a computer and/or a quiet place to study.

The advantage provided by this study is that of a larger dataset than has been the case for other studies for Ireland (excepting [8]) and critically the availability of a measure of mental health for the same people both before and after Covid. This is in contrast to other studies which have only followed mental health after the onset of the pandemic, and are unable to distinguish between individuals who were already experiencing depression before the pandemic and those whose depression arose following the onset of the pandemic. The richness of the dataset also permits investigation of whether the socioeconomic gradient of mental health issues differed pre and post pandemic and also able to examine transitions into and out of depression.

## Methods

### Study design

This study is a longitudinal, secondary analysis of quantitative data that were collected as part of the Growing Up in Ireland (GUI) study. The GUI is a longitudinal nationally-representative sample of children living in Ireland.

### Ethics

The same ethical protocols govern all waves of the GUI survey [9]. The relevant pieces of legislation are: (i) the Statistics Act, 1993 which provides a strong legal basis for the protection of all information collected in Growing Up in Ireland against unlawful disclosure; (ii) the Children First Act 2015 -which is designed to raise awareness of child abuse and neglect and to ensure an appropriate response to it; (iii) the Data Protection Acts 1988, 2003, 2018 which clarify the rights of persons with respect to personal data that is processed concerning them. The Irish Department of Children, Equality, Disability, Integration and Youth and the Irish Central Statistics Office are joint Data Controllers for the survey; (iv) the Data Protection Act 2018 (Section 36(2)) (Health Research) Regulations 2018 which provides for additional safeguards when processing data of a sensitive nature such as health data.

The following principles were also applied: (i) Informed consent, with the providing of information on the purpose of the study, the type of information gathered and what will be involved for participants (ii) Reporting concerns regarding risks to children with a protocol for reporting any incidents and for handling these appropriately (iii) Confidentiality of information provided, which is a legal requirement under the Statistics Act (iv) Avoidance of harm (including embarrassment/distress) (v) Instruments and protocols were reviewed by an independent Research Ethics Committee.

Written consent was obtained for all participants in the study. Assent was also sought from the children who participated. Re-analysis of the GUI dataset does not require additional ethical approval in accordance with the Central Statistics Office of Ireland who hold the GUI dataset.

## Study population

The data consist of two cohorts of Irish children and their PCs, the 2008 Cohort born in the period December 2007-June 2008 and the 1998 Cohort born in the period November 1997-October 1998 [10, 11]. The specific GUI data analysed is the last available (pre-Covid) waves of the GUI 2008 and 1998 Cohorts (collected in 2017/18 and 2018/19 respectively) and the Covid survey which was sent out in December 2020.

Online surveys, including questions on mental health, were carried out for both cohorts for most of the month of December 2020. The 2008 cohort survey began on December 4 and the 1998 cohort survey began on December 11, both surveys ending at the end of December 2020. The 1998 cohort was aged between 22 and 23 at the time of this survey. The 2008 cohort was aged 12 at the time of the survey and their PCs were also surveyed. This group ranged in age from 33 to 55. No data were collected for the PCs of the older cohort.

The latest pre-Covid wave of the 2008 Cohort consisted of 8032 children and their PCs and the original sampling frame used was the Child Benefit Register. This is a universal payment (made on behalf of all children regardless of socioeconomic status) and payment is made directly to the principal carer of the child (most typically the resident mother or step mother) and must be claimed within six months of the child being born, in the six months after the child becomes a member of the family or six months after the family become resident in Ireland. In the case of the 1998 Cohort the latest pre-Covid wave consisted of 5190 young adults and the original sample frame was the national primary school system, with 910 randomly selected schools participating in the study.

The following exclusions were placed on the data: for both cohorts only a balanced panel was used i.e. the study children (or young adults) and their PCs who responded to the Covid survey and the last pre-Covid survey. In addition, observations where the questions on mental health were not answered were also excluded. Finally, it was not possible to include the children from the 2008 cohort in the analysis as different mental health measures were collected in the Covid survey and in the last pre-Covid survey, and thus a comparison was not possible.

Following these exclusions, this left two study groups: the young adults from the 1998 Cohort (1953 observations, 1246 females and 707 males), and the PCs from the 2008 Cohort (3547 observations, 3457 females and 90 males). Since attrition from the GUI dataset is not random [12] sampling weights from the Covid survey were used in all the analysis.

## Measurements

The measure of mental health used for both of these study groups is the CES-D8 scale [13]. The original version of the CES-D scale had 20 items and has been used extensively across the

world and has featured in many published journal articles [14, 15]. There are also shorter versions of the measure which take less time to administer but are still regarded as reliable measures of depression. One of these is the CES-D8 and this is the version which is measured in GUI.

The CES-D8 measure consists of eight statements regarding how the respondent was feeling in the past week (e.g. "I felt depressed", "I felt fearful" etc). The respondent then indicates whether they experienced this feeling rarely/none of the time, some or a little of the time, occasionally or a moderate amount of the time or most or all of the time. Answers are coded 0, 1, 2 or 3 respectively, so that the minimum score possible is 0 and the maximum is 24. Higher scores indicate worse mental health and a score at or above 7 is regarded as indicating depression [16]. The GUI data is truncated at 13 (i.e. all CES-D8 scores greater than or equal to 13 are coded as 13) and hence this study mainly focuses upon rates of depression rather than actual CES-D8 scores.

As well as analysing how CES-D8 changes pre and post Covid the study also examines its relationship to socioeconomic status (SES). Two measures of SES are used. For the 1998 cohort of young adults the highest level of education attained by their principal carer is used. Education is broken down into four categories: (1) up to and including completion of lower secondary schooling (2) completion of all secondary schooling (3) obtaining a post-secondary school diploma or cert and (4) completion of third level education. Also included is a category where this data is missing. The highest level of education obtained by the young adults themselves is not used as many of the 1998 cohort will not have completed education, or may have had their education disrupted by Covid. For the PCs of the 2008 cohort their highest level of education obtained is used, with the same categories as above.

The second measure of SES used is equivalized after tax household income, as measured in the last pre-Covid wave of GUI as this data was not collected in the Covid survey. This is calculated via the answer to questions on total net household income from all members and sources after deductions for income tax and social insurance. If households cannot give an exact figure then they answer a sequence of questions where they are presented with cards where they select the range into which they believe that family income falls. The measure of SES gradient used for equivalized income is rank based and it seems reasonable to assume that even if households exact level of income is not measured with complete accuracy that the *ranking* of households by income will be less prone to error.

The analysis of the socioeconomic gradient of CES-D8 is limited to these two measures. While data on other household characteristics are available in GUI, these two measures have the great advantage of being measured with reasonable accuracy (particularly in the case of maternal education) and they also provide a clear ranking of SES.

## Statistical analysis

The following statistical analysis was carried out:

The distribution of CES-D8 scores and the fraction indicated as depressed are provided for both study groups. This analysis is stratified by gender for the 1998 cohort and by the SES measure of education for both study groups.

The Shapley decomposition is applied to the change in the fraction depressed [17]. Suppose the measure of depression is characterised as $D = D(\mu, L, CESD^*)$ where $\mu$ is the average level of CES-D8, $L$ is the Lorenz curve for the distribution of CES-D8 and $CESD^*$ is the critical depression threshold (note that the cumulative distribution function for CES-D8 will be completely characterised by its mean and Lorenz curve).

If subscripts "0" and "1" refer to the two time periods in question, then the change in depression over time $D_1 - D_0$ can be written as

$$D_1 - D_0 = F_1(CESD^*) - F_0(CESD^*) = D_1(\mu_1, L_1, CESD^*) - D_0(\mu_0, L_0, CESD^*)$$

where $F_i$ is the cumulative distribution function for period "i". This can then be decomposed into growth and redistribution effects denoted by $D(\mu_1, L_0, CESD^*) - D(\mu_0, L_0, CESD^*)$ and $D(\mu_1, L_1, CESD^*) - D(\mu_1, L_0, CESD^*)$ respectively.

However, as is the case with any path dependence type analysis, the choice of which configuration to use as the base period is arbitrary. The choices are (i) to calculate the CES-D8 growth effect with the initial CES-D8 distribution held constant, and to calculate the CES-8D distribution effect holding the mean CES-D8 at the final level or (ii) to calculate the CES-D8 growth effect with the final CES-D8 distribution held constant, and to calculate the CES-D8 distribution effect holding the mean CES-D8 at the initial level. The choice between approach (i) or (ii) is arbitrary but the precise decompositions will vary. The standard solution to this (as followed in [17] for example) is to take the average of the two effects. This provides a growth effect of

$$\frac{1}{2}\left[D(\mu_1, L_0, CESD^*) - D(\mu_0, L_0, CESD^*)\right] + \frac{1}{2}\left[D(\mu_1, L_1, CESD^*) - D(\mu_0, L_1, CESD^*)\right]$$

and a redistribution effect of

$$\frac{1}{2}\left[D(\mu_0, L_1, CESD^*) - D(\mu_0, L_0, CESD^*)\right] + \frac{1}{2}\left[D(\mu_1, L_1, CESD^*) - D(\mu_1, L_0, CESD^*)\right]$$

These two expressions are the growth and distribution components for a two-way Shapley decomposition of the change in the rate of depression. The Shapley decomposition arises from the classic co-operative game theory problem of dividing a pie fairly. The solution is that each player is assigned her marginal contribution averaged over all possible coalitions of agents. The interpretation here was to consider the various n factors which contribute together to determine the change in the value of an indicator such as depression and then assign to each factor the average marginal contributions taken over the n! possible ways in which the factors may be removed in sequence. Since there are two factors (n = 2, growth and distribution) there are 2! = 2 possible routes. The decomposition is always exact as the factors are treated symmetrically.

To investigate the gradient of mental health/depression with respect to SES, how CES-D8 scores and depression rates vary by education of principal carer is examined. To examine the gradient with respect to equivalised income the concentration index is used [18]. Assume that the CES-D measure for individual i is given by $CESD_i$. Then if $r_i$ is the fractional rank of individual i in the income distribution (or whatever ranking variable is being used), the concentration index for CES-D is given by $C_{CESD} = \frac{2\,cov(CESD_i, r_i)}{\overline{CESD}}$ where $\overline{CESD}$ is the mean value of CES-D. $C_{CESD}$ can take on a value from -1 to +1, where a negative (positive) value indicates that CES-D takes on higher values among the relatively poor (rich). In the case of a binary measure (e.g. depressed/non-depressed) a normalisation must be applied to the index (since the bounds would not be -1 and +1). [19] suggested a normalisation of $C_{CESD,N} = 4\overline{CESD}\,C_{CESD}$, which is applied here.

The generalised concentration index is also calculated, which is simply the concentration index multiplied by the mean value of CES-D, $GC_{CESD} = \overline{CESD}\frac{2cov(CESD_i, r_i)}{\overline{CESD}} = 2cov(CESD_i, r_i)$.

**Table 1. Sample characteristics by cohort, gender, maternal education and CES-D8 scores pre and post Covid.**

| | 1998 Cohort | | | | Principal Carers 2008 Cohort (N = 3547) | |
|---|---|---|---|---|---|---|
| | Females (N = 1246) | | Males (N = 707) | | | |
| Principal Carer's Education | Pre Covid | Post Covid | Pre Covid | Post Covid | Pre Covid | Post Covid |
| Lower Secondary | 0.222 | | 0.204 | | 0.111 | |
| Complete Secondary | 0.390 | | 0.370 | | 0.355 | |
| Diploma/Cert | 0.157 | | 0.178 | | 0.221 | |
| Third Level | 0.166 | | 0.182 | | 0.313 | |
| Education missing | 0.064 | | 0.067 | | 0.000 | |
| | Females | | Males | | | |
| | Pre Covid | Post Covid | Pre Covid | Post Covid | Pre Covid | Post Covid |
| CES-D8 (mean) | 4.984 | 7.561 | 3.755 | 5.828 | 2.321 | 3.978 |
| Proportion Depressed | 0.307 | 0.558 | 0.225 | 0.403 | 0.107 | 0.240 |

Finally, transition matrices are calculated for both study groups and in the case of the 1998 cohort the analysis is stratified by gender. These matrices are calculated both for the depression rate and also for the more granular CES-D8 scores.

## Results

The summary statistics presented in Table 1 show the background of both study groups in terms of education (education of principal carer in the case of the 1998 cohort) and how mental health changed pre and post Covid. The 2008 cohort showed higher education generally (reflecting the fact that the principal carers are a younger cohort), and also had no "missing" responses. For both genders and both of the study groups a deterioration in mental health is observed post Covid. The 1998 cohort shows nearly a doubling of the fraction with a CES-D8 score in excess of the depression threshold and also shows a higher base rate for females, with a majority of females indicating depression. The fraction of principal carers from the 2008 cohort reporting depression more than doubled, but from a considerably lower base. This data is also summarised in Figs 1 and 2.

### Shapley decomposition

As shown in Table 2, for the 1998 cohort, the bulk of the change in the rate of depression is accounted for by a general growth in the level of CES-D8 and this is particularly so for males. For females, a change in the distribution accounts for about 20% of the increase in depression, but overall the increase in depression over the period is mostly accounted for by a general rise in CES-D8 rather than by any changes in the shape of the distribution. Note that owing to the top-censoring of the distribution at 13, the calculated mean CES-D8 is underestimated both pre and post pandemic. This will only affect the breakdown of the change in the fraction depressed if we believe that the average CES-D8 for those above 13 is *higher* post rather than pre pandemic. This seems plausible, in which case the percentage of the change in the rate of depression accounted for by overall growth as opposed to redistribution is a *lower* bound.

For the principal carers of the 2008 cohort the growth in the *level* of CES-D8 over-explains the total change. Or to phrase it another way, if only the change in the distribution of CES-D8 pre and post pandemic was observed, while holding the mean constant, then depression would have *fallen*. Thus the rise in depression was completely caused by an overall deterioration in

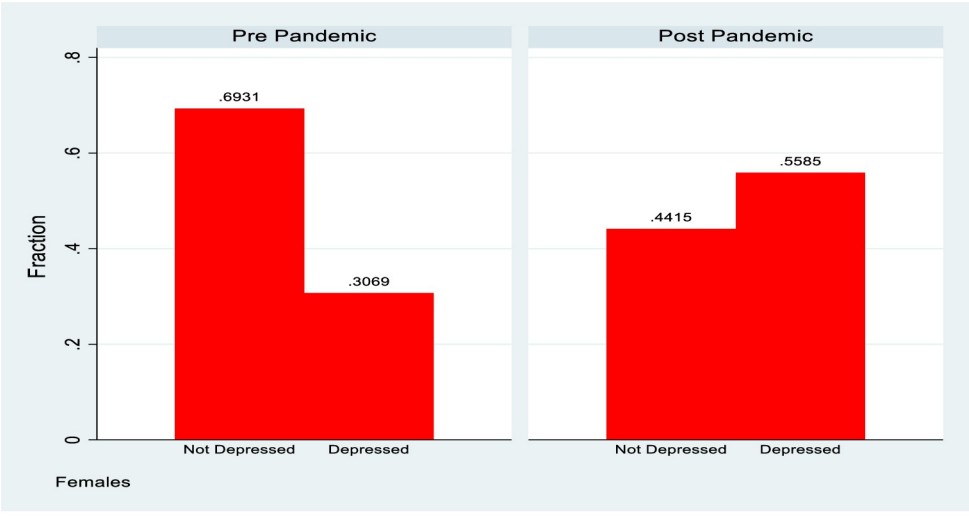

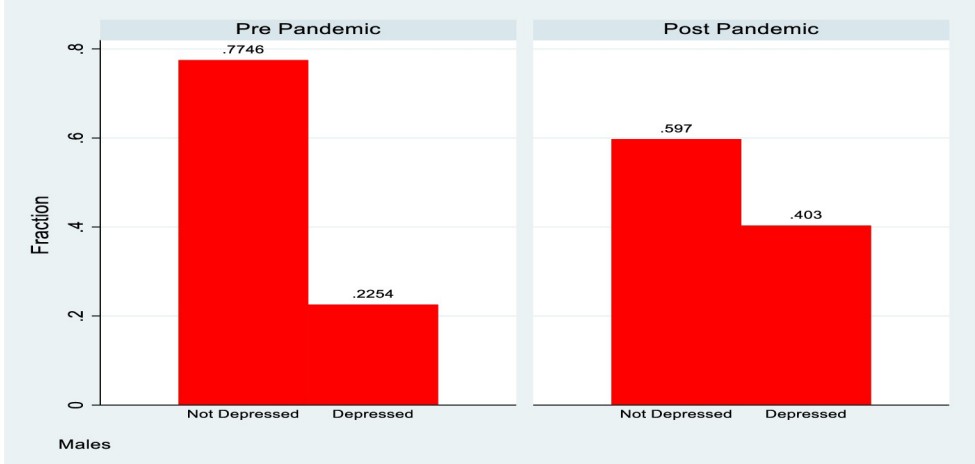

**Fig 1. Depression rates, 1998 cohort, by gender.**

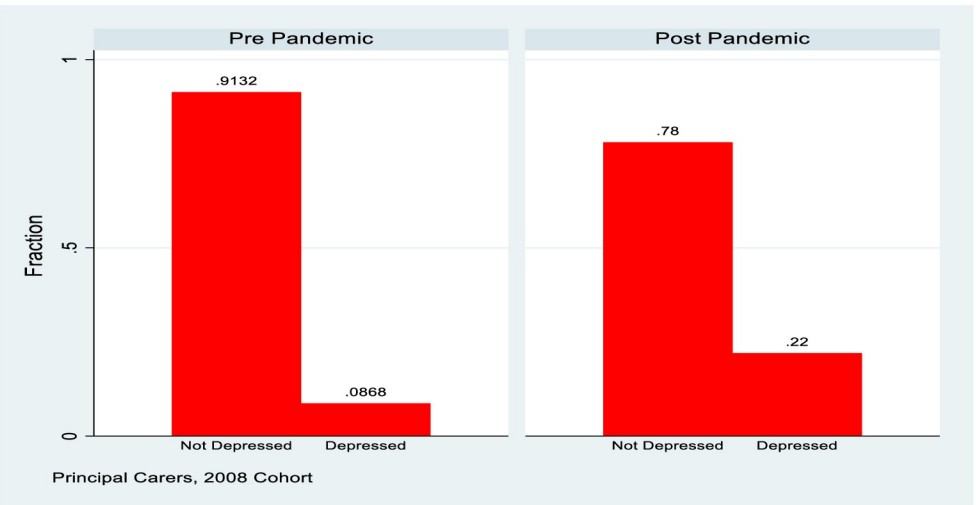

**Fig 2. Depression rates, 2008 cohort, principal carers.**

**Table 2. Shapley decomposition for change in depression, pre and post pandemic.**

| Measure | Fraction Depression, CES-D ≥7 | | | | |
|---|---|---|---|---|---|
|  | Pre | Post | Change | Growth Contribution (%) | Distribution Contribution (%) |
| **Total (1998)** | 0.266 | 0.481 | 0.215 | 0.172 <br> 80% | 0.043 <br> 20% |
| **Female (1998)** | 0.307 | 0.559 | 0.252 | 0.18 <br> 71.6% | 0.07 <br> 21.4% |
| **Male (1998)** | 0.225 | 0.403 | 0.178 | 0.164 <br> 92.1% | 0.014 <br> 7.9% |
| **Principal Carers (2008)** | 0.107 | 0.240 | 0.133 | 0.157 <br> 117.8% | -0.024 <br> -17.8% |

CES-D8 scores for everyone, rather than a spread in the distribution which pushed some people over the threshold.

## Socioeconomic gradient by maternal education

Table 3 shows that for the 1998 cohort, the depression rates by education of PCs for females show little variation, and if anything a higher rate for those with higher PC education post covid. The figures for males are more mixed. Pre-covid, the lowest depression rates were observed for those with highest PC education, but the relationship was not monotonic. Those with the lowest PC education did not show the highest rates of depression. In the post-covid situation there is very little discernible gradient and depression rates for those with highest PC education have caught up with the average for males.

For the PCs of the 2008 cohort depression rates are highest for those with the lowest education level, followed by those who have completed secondary school education. The rates for those with a diploma or cert or third level education are very close. A standard difference in proportions test show a statistically significant increase post pandemic for all groups and the absolute differences are highest for those with the two lower levels of education. Relative rates of increase however are greater for the higher levels of education, so in relative terms at least, there is some sign of convergence in rates of depression, albeit at a higher level. This information is also captured in Figs 3–5.

## Socioeconomic gradient by income

For the 1998 cohort, Table 4 shows that while the concentration index is negative for females and positive for males, in all cases it is insignificantly different from zero, indicating no socioeconomic gradient to either CES-D8 or to depression, in total, or by gender.

For the 2008 cohort of PC, the concentration index shows a fall post pandemic. However, the *generalised* concentration index i.e. the concentration index multiplied by the average level of CES-D8 shows an increase. The concentration index for the rate of depression also rises. Thus in terms of how the socioeconomic gradient has changed, the picture is quite complex. In purely relative terms the gradient is less steep post-pandemic. However allowing for the change in the average level of CES-D8 the gradient is steeper reflecting a greater concentration of higher CES-D8 values amongst the poorer and less well-educated, and this is also confirmed by the results in Table 3.

## Transitions in mental health pre and post Covid

As well as looking at overall rates of depression, mobility over the course of the pandemic is examined using transition matrices and a graphical version is presented using the *tabplot*

**Table 3. Change in rate of depression by education, standard errors in brackets, *** p<0.001, **p < .01.**

| 1998 Cohort | Depression Rate Pre Covid | Depression Rate Post Covid | Difference |
|---|---|---|---|
| **Females** | | | |
| *Lower Secondary* | 0.309 | 0.524 | 0.216*** (0.055) |
| *Complete Secondary* | 0.307 | 0.576 | 0.269*** (0.039) |
| *Diploma/Cert* | 0.316 | 0.574 | 0.258*** (0.047) |
| *Third Level* | 0.305 | 0.558 | 0.253*** (0.041) |
| *Missing* | 0.280 | 0.535 | 0.255** (0.103) |
| **Males** | | | |
| *Lower Secondary* | 0.219 | 0.394 | 0.175** (0.081) |
| *Complete Secondary* | 0.261 | 0.417 | 0.156*** (0.034) |
| *Diploma/Cert* | 0.236 | 0.354 | 0.118** (0.055) |
| *Third Level* | 0.167 | 0.426 | 0.259*** (0.045) |
| *Missing* | 0.178 | 0.422 | 0.245** (0.104) |
| **TOTAL** | **0.266** | **0.481** | **0.215*** (0.072)** |
| **2008 Cohort (Principal Carers)** | **Depression Rate Pre Covid** | **Depression Rate Post Covid** | **Difference** |
| *Lower Secondary* | 0.2188 | 0.3618 | 0.1430*** (0.0509) |
| *Complete Secondary* | 0.1144 | 0.2705 | 0.1561*** (0.0202) |
| *Diploma/Cert* | 0.0852 | 0.1973 | 0.1120*** (0.0182) |
| *Third Level* | 0.0747 | 0.1923 | 0.1176*** (0.0125) |
| **TOTAL** | **0.1071** | **0.2399** | **0.1329*** (0.0107)** |

command in Stata, which combines the transition matrix with a histogram. The underlying data used to construct it can be obtained from a cross tabulation of the data by CES-D8 score and GUI wave. In the case of the matrix for CES-D8 scores it is a 14x14 matrix, where the column shows the fraction of the sample with that CES-D8 score pre-Covid, and the row indicates the fraction of the sample with that CES-D8 score post-Covid. The entries along the main diagonal of the matrix indicate the fraction whose CES-D8 score does not change pre and post Covid. This is presented in Fig 6 for the 22/23 year olds from the 1998 cohort, where depression is defined as a CES-D8 score greater than or equal to 7.

Fig 6 shows that of the 73.3 per cent of the sample (45.7 per cent + 27.6 per cent) of those who were not depressed pre-Covid, 27.6 percent transitioned into depression post-Covid. Of the 26.7 per cent who were depressed pre-Covid, 6.2 per cent transitioned back into non-depression but 20.5 per cent remained depressed. Fig 7 shows higher percentages of females depressed pre Covid and then also a higher rate of transition into depression post Covid. Over 31 per cent of females become depressed post Covid as opposed to 24 per cent of males.

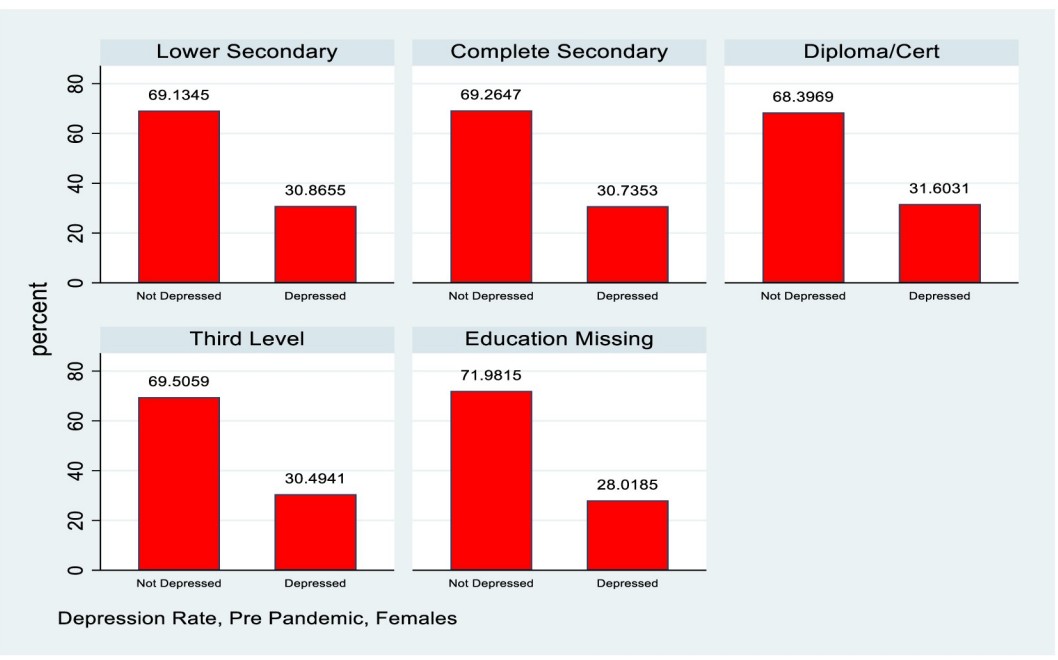

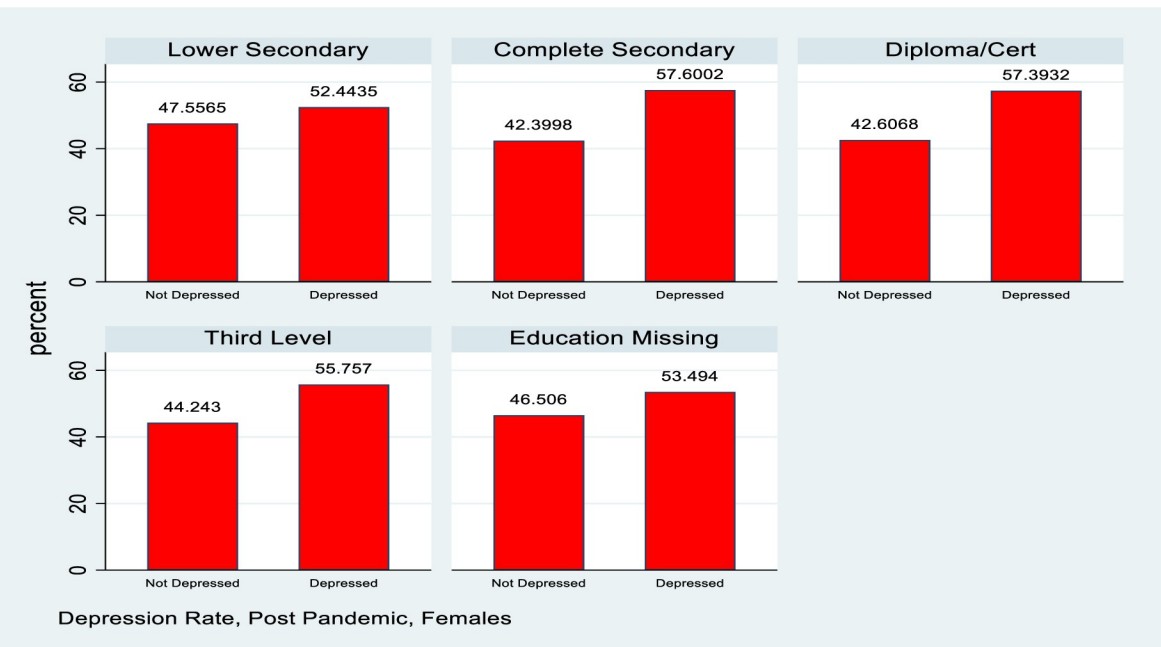

**Fig 3. Depression rates pre and post pandemic, 1998 cohort, females, by principal carer education.**

Fig 8 shows the equivalent graphs for PCs from the 2008 cohort. What is perhaps most notable here is the relatively low rate of transition out of depression over the pandemic, in contrast to the 1998 cohort, perhaps reflecting lesser volatility of mental health.

It is also possible to examine mobility using these transition matrices at a more granular level. The matrix in Fig 9 shows transitions across different values of CES-D8 for the 1998 cohort. In examining these graphs there are a number of factors to bear in mind. Transitions

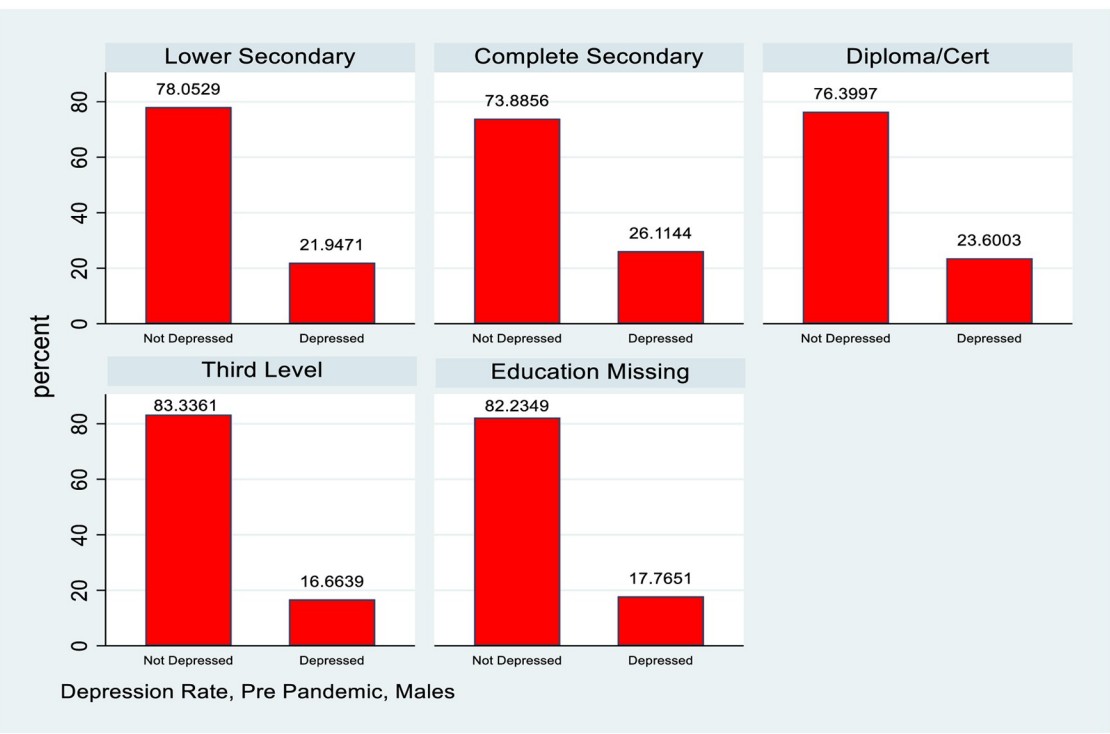

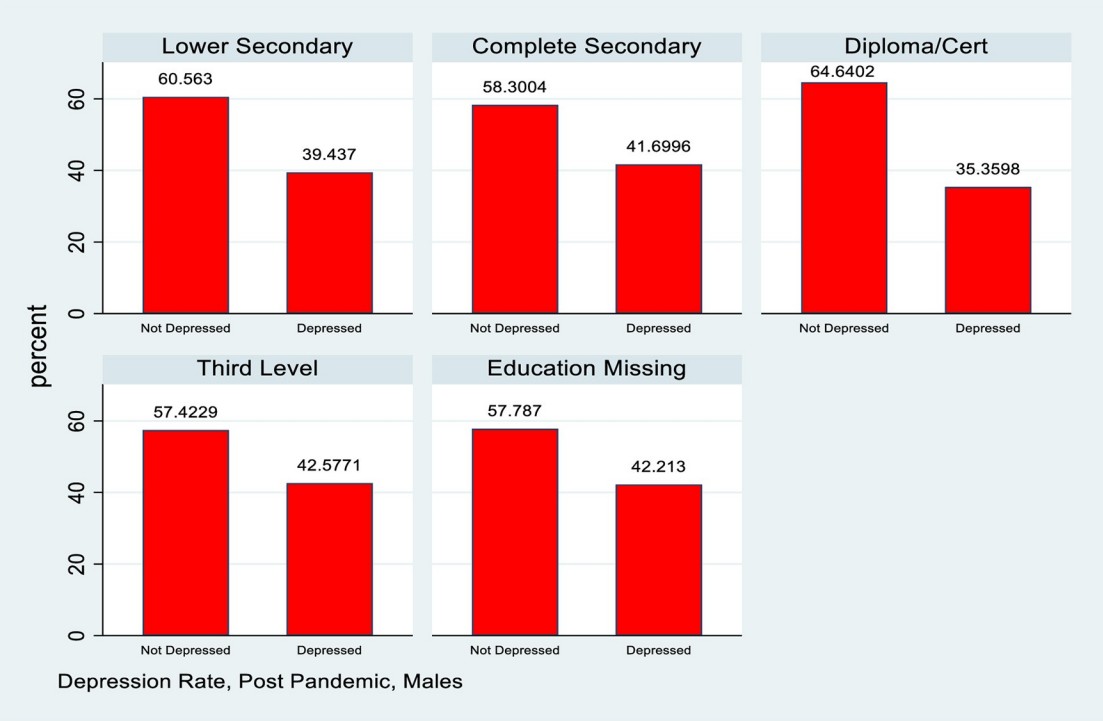

**Fig 4. Depression rates pre and post pandemic, 1998 cohort, males, by principal carer education.**

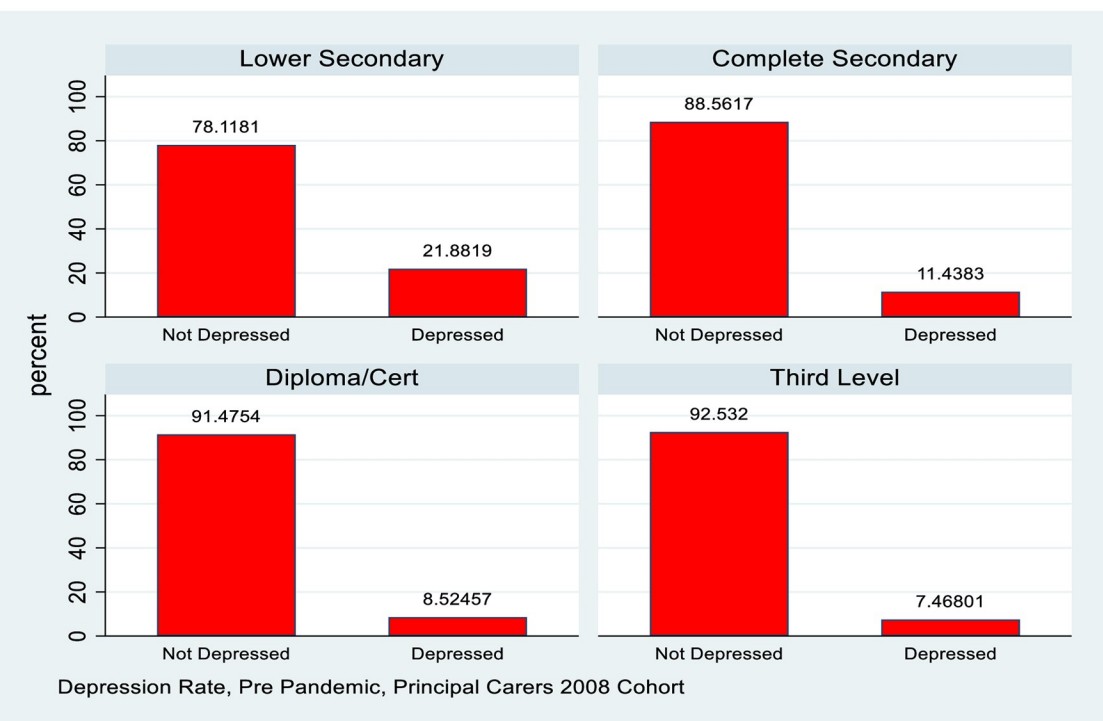

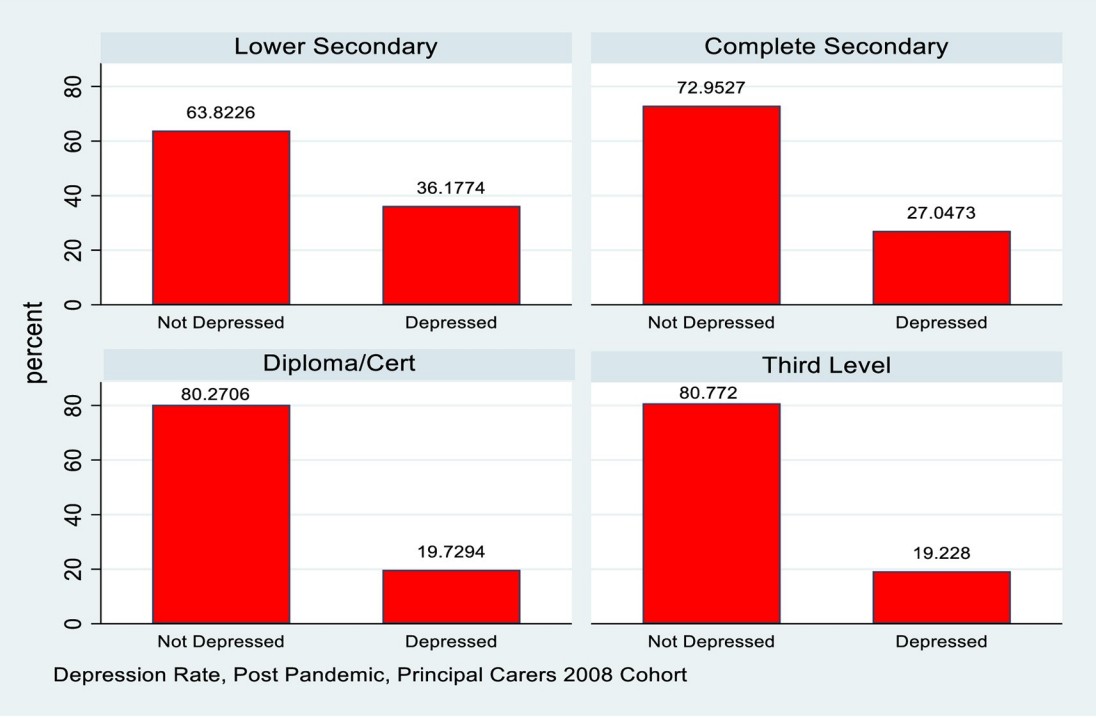

**Fig 5. Depression rates pre and post pandemic, 2008 cohort, principal carers, by education.**

**Table 4. Concentration and generalised concentration indices for CES-D8 and depression, standard errors in brackets, *** p<0.001, **p < .01.**

| | CES-D, Concentration Index | | CES-D, Generalised Concentration Index | | Depression, Concentration Index | |
|---|---|---|---|---|---|---|
| | Pre | Post | Pre | Post | Pre | Post |
| **2008 Cohort** | | | | | | |
| *Females* | -0.026 (0.022) | -0.018 (0.015) | -0.128 (0.109) | -0.139 (0.112) | -0.037 (0.045) | -0.049 (0.049) |
| *Males* | 0.038 (0.029) | 0.009 (0.022) | 0.143 (0.110) | 0.053 (0.128) | 0.020 (0.048) | 0.055 (0.057) |
| **Total** | -0.003 (0.013) | -0.011 (0.013) | -0.015 (0.080) | -0.072 (0.087) | -0.015 (0.033) | -0.007 (0.038) |
| **1998 Cohort** | | | | | | |
| **Total** | -0.107*** (0.021) | -0.069*** (0.014) | -0.248*** (0.049) | -0.275*** (0.053) | -0.106*** (0.020) | -0.137*** (0.025) |

which happen below the critical value of 7 are arguably of less interest since they do not involve a transition across the depression threshold. Because the data are truncated at values of 13 we do not observe transitions from say a value of 14 to 17, they are all absorbed in the category ">= 13". Finally, no cardinality can be assigned to these transitions e.g. it is not possible to say that a move from 2 to 4 represents the same change as a move from 4 to 6, or from 8 to 10. Nevertheless, it seems reasonable to suggest that a move from 2 to 12 represents a greater deterioration in health than a transition from 2 to 8, or from 1 to 8.

The matrix is divided into four quadrants. The north-west quadrant represents people who are not depressed either pre or post Covid, so in some sense they are of less concern. While a move to the right within this quadrant is better avoided as it indicates a deterioration in mental health, since it does not involve a move across the threshold it is arguably of less concern than moves which *do* cross the threshold.

The south-west quadrant is the quadrant of good news as it comprises people whose mental health improved over the pandemic and who moved from depressed to non-depressed. However, there are relatively few of those. The north-east and south-east quadrants are of most

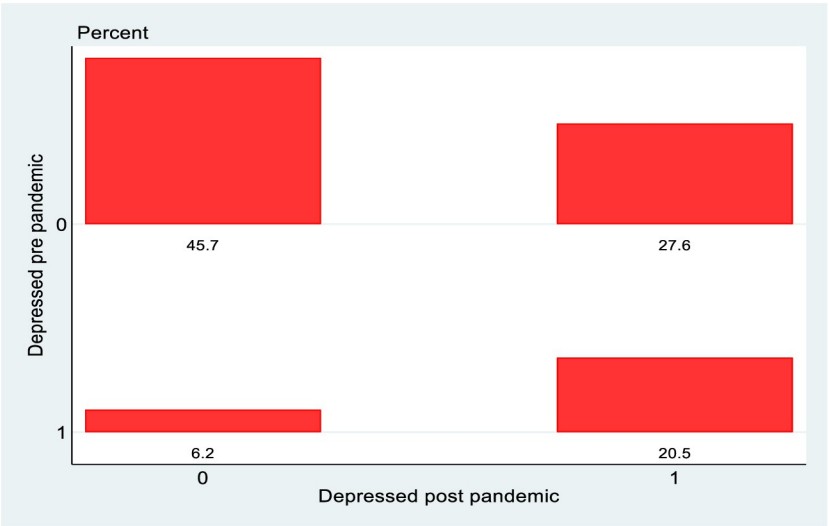

**Fig 6. Transition matrix, 1998 cohort, total.**

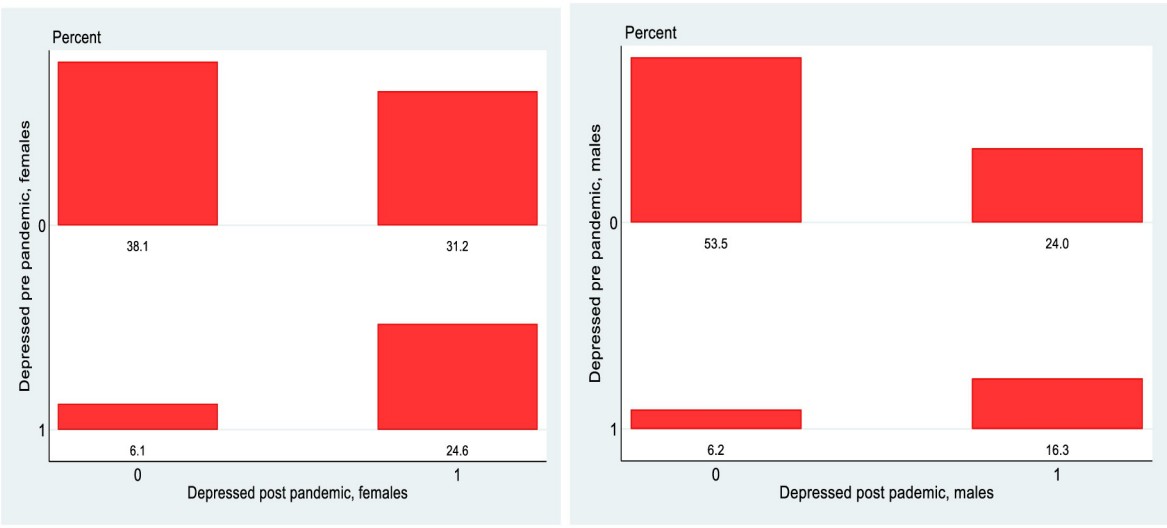

**Fig 7. Transition matrix, 1998 cohort, by gender.**

concern, and the advantage of the detailed transition matrix is that it does not just tell us that people crossed into depression (or stayed depressed) but it also gives some idea (allowing for the caveat re cardinalisation mentioned above) of *how much* someone's mental health deteriorated. Given this, it is notable that if looking at all the columns in the north-east quadrant, the most heavily populated is the right-most one, where the post-Covid CES-D8 score is highest. In other words, for those who transitioned into depression, excepting those who had a pre-Covid CES-D score of 0 or 1, for all the others a higher proportion moved to a post-Covid score of 13 or over, the highest (worst) score achievable, than any other score. Mental health did not just deteriorate, it seems to have deteriorated quite significantly.

Finally, in the south-east quadrant of people who were depressed pre and post Covid it is also noticeable that the right-most column is the most heavily populated. People whose mental

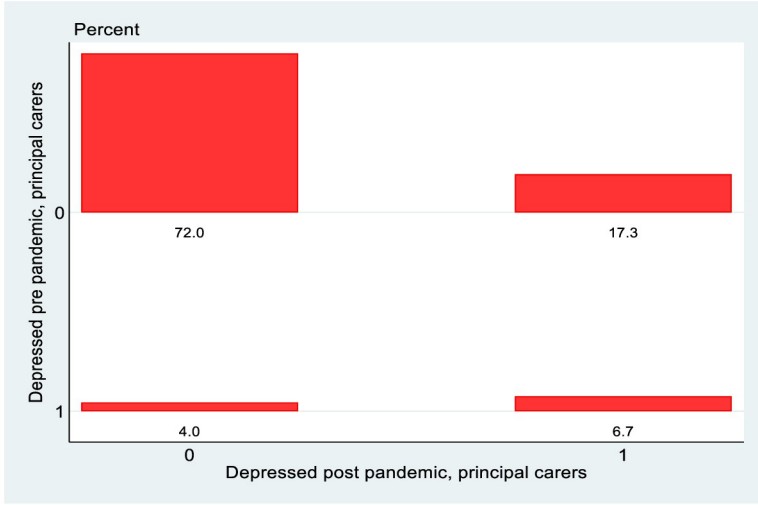

**Fig 8. Transition matrix, 2008 cohort, principal carers.**

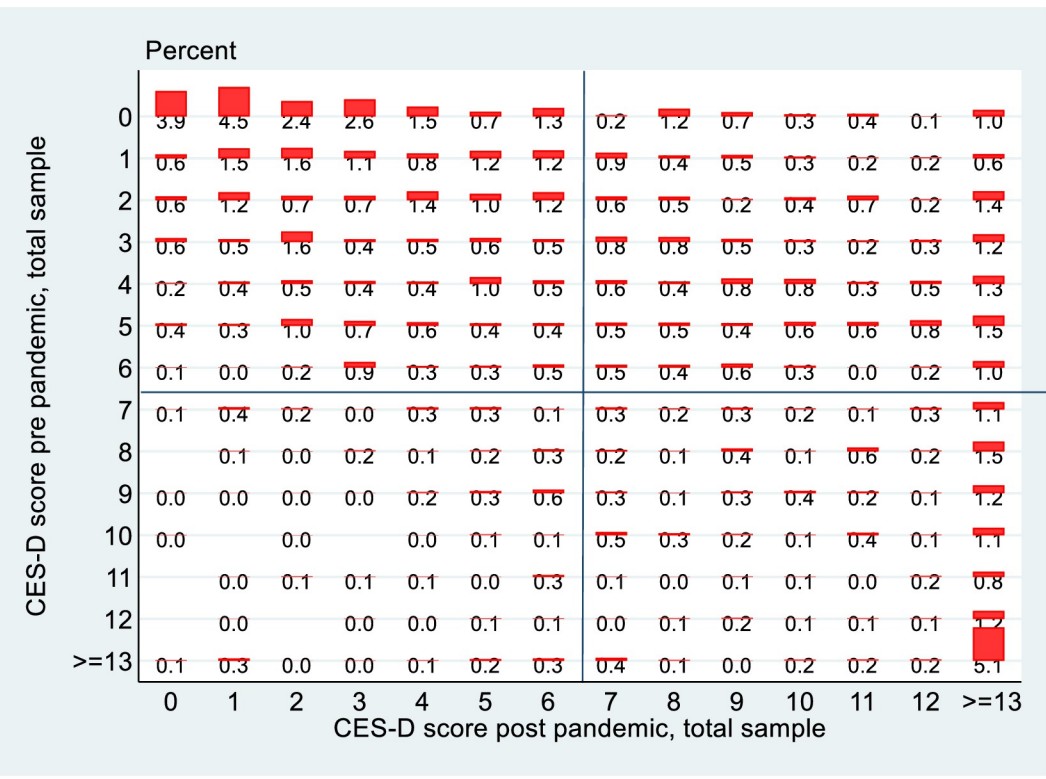

**Fig 9. Transition matrix, 1998 cohort, total sample, detailed.**

health was above the depression threshold saw further deterioration, in many cases to the highest (worst) recorded CES-D8 score.

The detailed transition matrices by gender are shown in Fig 10. What is probably most noticeable about these matrices is that the phenomenon of moving to the worst state is clearly more pronounced for females than for males. To use an informal metric, of the columns in the north-east quadrant for females in 6/7 cases the most populated is the right-most one, while for males it is only 3/7. There was a deterioration in mental health for males but it seems to have been much less severe than was the case for females.

Finally Fig 11 shows the same detailed transition matrix for PCs of the 2008 cohort. As is already evident in the less detailed matrix looking just at depression, the overall level of depression and of transition into depression is less than is the case for the young adults of the 1998 cohort. The "heavy rightmost column" phenomenon in the north-east quadrant also does not seem to be present. In no case is the right-most column of the north-east quadrant the most heavily populated. While the mental health of the PCs of the 2008 cohort did deteriorate, the deterioration seems to have been less severe than in the case of the young adults of the 1998 cohort.

## Discussion

This study examines the evolution of mental health in Ireland pre and post the Covid 19 pandemic for two distinct groups. First of all a cohort of young adults born in 1997–98 and secondly a group of PCs (predominantly female) of a cohort of children born in October 2008. A particular contribution of this study is the use of a consistent measure of mental health (the

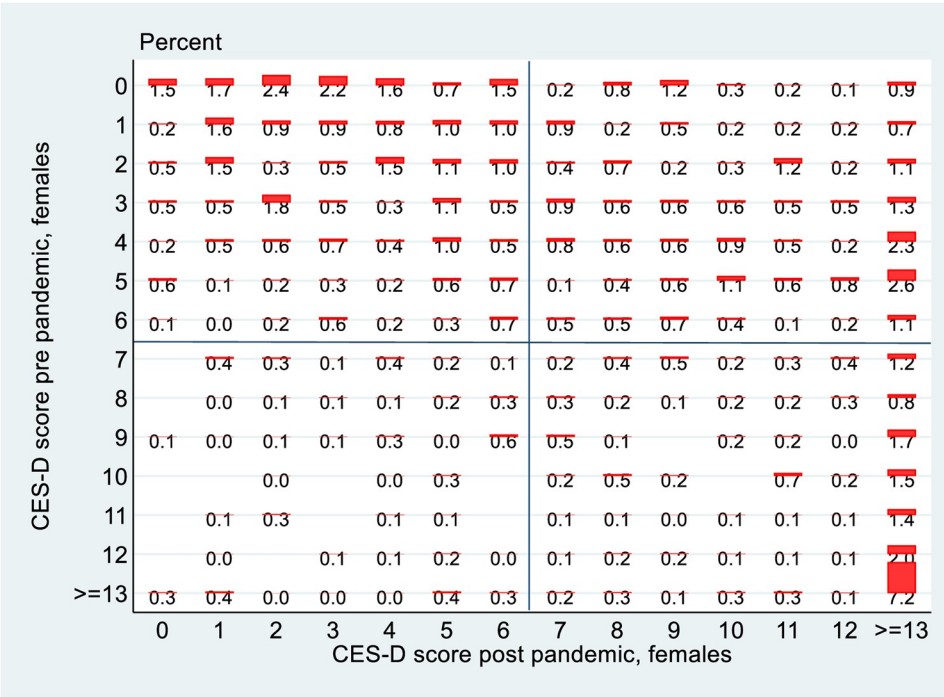

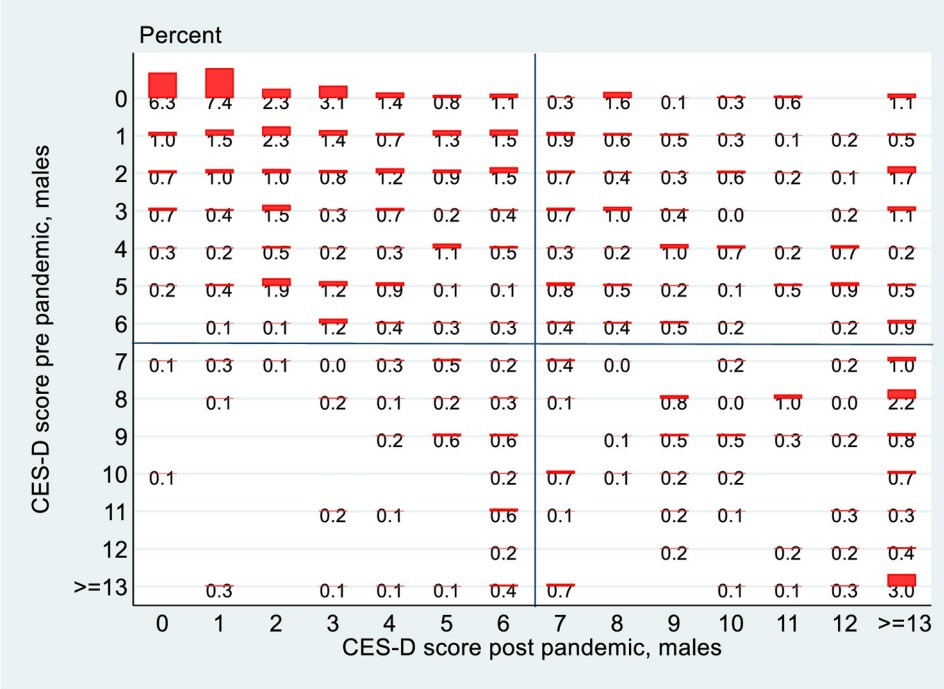

**Fig 10. Transition matrices, 1998 cohort, detailed, by gender.**

CES-D8) before and after the pandemic for the same group of people, thus controlling for time invariant characteristics.

The study finds that mental health deteriorated for both these groups. The deterioration was most pronounced among females of the 1998 cohort. Their CES-D8 scores pre-pandemic

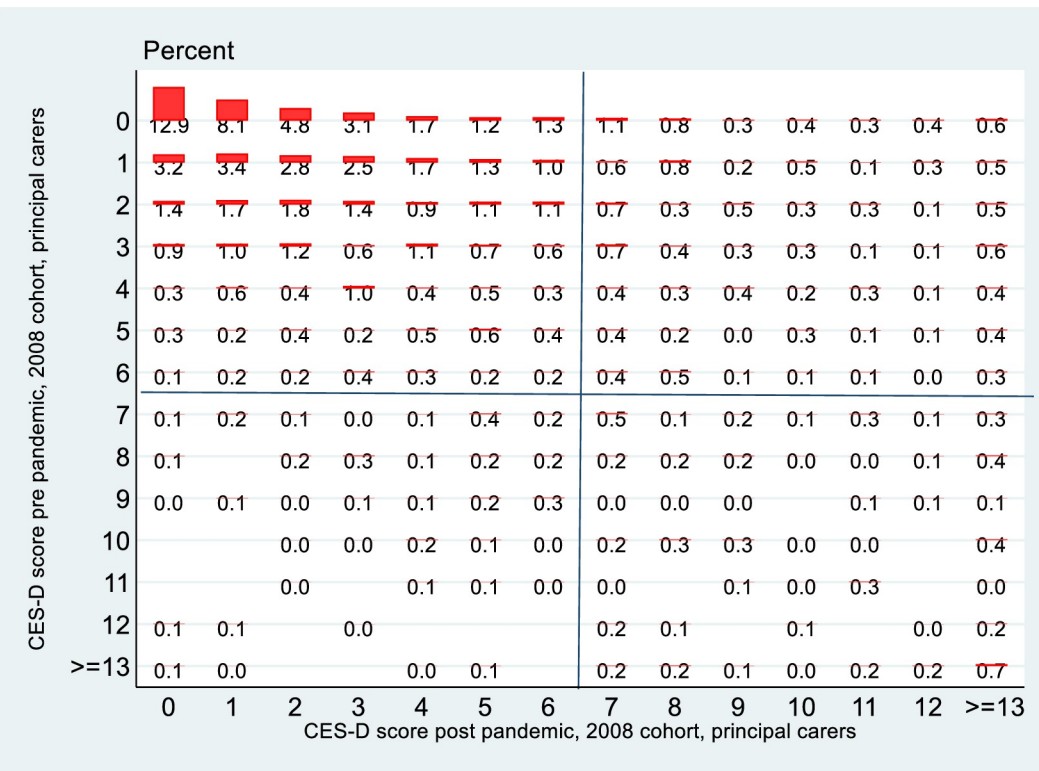

**Fig 11. Transition matrix, 2008 cohort, principal carers, detailed.**

were the highest (indicating worse mental health) and they also showed the greatest deterioration post pandemic, to such an extent that the majority of females of the 1998 cohort had a score at or above the key threshold of 7, thus indicating depression. Depression rates for males from the 1998 cohort and for the principal carers of the 2008 cohort also showed increases, though the level of depression for the latter group was considerably lower, at around 22 per cent. This difference by gender is consistent with existing literature in this area [20].

The analysis also sheds light on how the rate of depression (as measured by the fraction of the sample with a CES-D8 score greater than or equal to 7) evolved. Effectively this involves looking at how the mass of the distribution to the right of a specific threshold changes over time. This can happen because the whole distribution shifts to the right (the "growth effect") while retaining the same shape, or because the shape of the distribution changes (the "distribution effect") or both. The results show that for both males and females from the 1998 cohort that the growth effect dominated, accounting for between 70 and 90 per cent of the overall increase. Thus the increase in the rate of depression was mostly caused by an overall decline in mental health, rather than a decline for those near the threshold. The dominance of the growth effect is even greater for the principal carers of the 2008 cohort, where it accounts for more than 100 per cent of the change.

The analysis also examines how the distribution of mental health with respect to measures of socioeconomic status (SES) evolved pre and post pandemic. For many health conditions a "gradient" is observed where those with lower SES (whether measured by income, social class or education) have poorer health outcomes [18]. This study uses household after tax income and PC education as measures of SES for the 1998 cohort and actual educational attainment and after tax household as measures for the PCs of the 2008 cohort. Figs 3 and 4 show how

depression rates change by PC education for the 1998 cohort and Fig 5 shows this information by education for the PCs of the 2008 cohort, with the underlying numbers in Table 3.

For females in the 1998 cohort there is little evidence of a gradient in depression by PC education either pre or post pandemic, though of course the overall rate of depression has increased. For males there is some evidence of a slight gradient pre pandemic with depression rates lower for those whose PCs have post secondary school education, but this has mostly vanished post pandemic. Again, the overall rate of depression increased but it is noticeable that there is very little evidence of a gradient by SES, and, whatever slight one did exist, disappeared with the pandemic.

Fig 5 shows a discernible gradient in depression rates by education both pre and post pandemic for the PCs of the 2008 cohort. The relative gaps by education compress slightly post pandemic, but since absolute levels are considerably higher the absolute gaps have widened.

Results are also presented for these gradients by income and in this case, given that income is a continuous cardinal variable it is possible to calculate a summary measure, the concentration index. Clearly income and education are not equivalent but they do tend to be quite highly correlated so it is no surprise to see that the results by education are broadly replicated. For the 1998 cohort there is no statistically significant gradient for either gender, pre or post pandemic. For the PCs of the 2008 cohort, when measured on a purely relative basis a slight fall in the concentration index is observed, but when allowing for the increase in the overall rate of depression, the index increases slightly. Overall, however, it seems fairest to say that the pandemic had little effect on what SES gradients in mental health already existed pre pandemic and this finding is relatively unusual, compared to other health conditions.

The final part of the analysis specifically looks at mobility, exploiting the longitudinal nature of the data. Again, it is instructive to stratify the analysis by gender for the 1998 cohort. Reflecting the changes in the overall rates of depression, there are more entries into depression rather than exits from depression and this is more pronounced for females than males in the 1998 cohort, which in turn is more pronounced than for the PCs of the 2008 cohort. Perhaps the greatest difference is seen however when we look at the more granular transition matrices in Figs 10 and 11. The data on CES-D8 is truncated at a value of 13, so transitions to the very highest values (and hence worst mental health outcomes) cannot be observed. However CES-D8 values of 13 or greater clearly indicate more severe depression than values just above the threshold of 7. In this regard it is striking that of those females from the 1998 cohort who transition into depression, a noticeably higher fraction record values of 13 or greater than is the case for males or for the PCs from the 2008 cohort. Thus it seems fair to say that for the three distinct groups observed in this study, not only did females from the 1998 cohort see a greater increase in the rate of depression over the pandemic, but they also seem to have moved to a more intense level of depression. Again, this is consistent with the studies such as [21] who used longitudinal Dutch data to show a greater increase in depression for females following lockdown, although interestingly they showed men experiencing more anxiety. Unfortunately, the data in GUI deals with depression rather than anxiety so it is not possible to investigate whether anxiety levels increase more for men in Ireland during lockdown.

In terms of how this study differs from other studies in the area, it is perhaps the only one (apart from [8]) which explicitly examines the same group of people before and after Covid. It also differs from other studies in specifically examining the dynamics of mental health via the use of transition matrices and is able to distinguish between growth and distribution effects. Finally, it also formally analyses the socioeconomic gradient of CES-D8 scores pre and post pandemic and finds a limited role for what measures of SES are available in GUI.

However, it is also important to acknowledge some potential caveats with the analysis. Thus it is possible that the increase in depression for the 1998 cohort over the years leading up to

December 2020 could be to some degree explained by age. As [3] show in their study of mental health in the UK during the early stages of the pandemic, well-being tends to show a U shaped relationship with age, with fairly steep changes in mental health problems for younger people, and also for those entering retirement. They also show that there is a seasonal aspect to mental health, and that it tends to deteriorate going into winter months. The GUI Covid surveys were held in December 2020 and evidence from the UK Household Longitudinal Study (UKHLS) suggests that seasonal factors are significant for this month. Banks and Xu use the rich data available in the UKHLS to estimate a counterfactual of how mental health might have evolved in the absence of the pandemic and their analysis for the group comparable in age to the 1998 cohort suggests that the pandemic accounted for about 75% of the observed deterioration in mental health, with age-related developments accounting for the remainder. Unfortunately GUI lacks a consistent measure of mental health over a sufficient sequence of waves to permit the construction of a similar counterfactual, but this issue should be borne in mind when interpreting the results in this study. The lack of information on the month of interview for the pre pandemic survey also prevents from taking account of potential seasonal factors.

In terms of the mobility analysis, since only two data points are observed (the surveys pre and post pandemic) it is possible that transitions into and out of mental health which happen *between* these surveys are not observed. Thus a transition into (or out of) depression may have occurred after the pre pandemic survey but *before* the arrival of Covid, perhaps for the age related reasons referred to above. It is also possible that an observation may be observed to have no transition yet may have transferred for example into and back out of depression.

In terms of what policy conclusions can be drawn from this study, it must be borne in mind that the Covid 19 pandemic was an event without precedent for much of the world. While it is certainly possible, if not likely, that there will be future pandemics, their precise nature may well differ from Covid and the policy responses, certainly in terms of responses likely to have an impact upon mental health, may also differ. However, regardless of the precise nature of the next pandemic, it would be useful to have a record of mental health and well-being *before* the pandemic i.e. it is something which should be monitored on a regular basis. Then, should a pandemic arise and should there be policies which might affect mental health, then it will be possible to analyse before and after effects. In the Irish case it would also be useful to have this data across a broader sample of the population, rather than those parts merely covered by GUI. Finally, the role of gender in the analysis should also be stressed. Young females were more vulnerable to deterioration in mental health and this was a cohort who were already be at a disadvantage. Care should be taken to monitor the health and well-being of this group in particular. To summarise, one of the main implications and recommendations from this study concerns the type of information which public health authorities should be collecting routinely and which then can be called upon should another pandemic arise.

## Conclusion

This study examines the evolution of mental health as measured in December 2020, nine months into the pandemic compared to observations pre pandemic for two cohorts of people. A deterioration in mental health was observed for both cohorts and particularly for younger women. The increase in the rate of depression predominantly occurred due to an overall decline in mental health rather than being concentrated amongst those already vulnerable (in the sense of being near the depression threshold). There was little, if any, change in the socio-economic gradient associated with mental health and virtually no gradient at all was observed pre or post pandemic for the 1998 cohort. Finally, mobility analysis revealed that not only did females from the 1998 cohort show greater transitions into depression, they also appeared to

transition into more extreme levels of depression. This may be partly due to age related changes which would have occurred anyway in the absence of the pandemic. It remains to be seen what at what level mental health will eventually stabilise post pandemic.

## Acknowledgments

I would like to acknowledge helpful comments from Orla Doyle and participants at an internal seminar in University College Dublin. The usual disclaimer applies.

## Author Contributions

**Conceptualization:** David Madden.

**Formal analysis:** David Madden.

**Investigation:** David Madden.

**Methodology:** David Madden.

**Writing – original draft:** David Madden.

**Writing – review & editing:** David Madden.

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
