## [Decision Letter · Decision Letter 0]

5 Jul 2023

PONE-D-22-33706Mental Health in Ireland During the Covid Pandemic: Evidence from Two Longitudinal SurveysPLOS ONE

Dear Dr. Madden,

Thank you for submitting your manuscript to PLOS ONE. After careful consideration, we feel that it has merit but does not fully meet PLOS ONE’s publication criteria as it currently stands. Therefore, we invite you to submit a revised version of the manuscript that addresses the points raised during the review process.

We look forward to receiving your revised manuscript.

Kind regards,

Enamul Kabir

Academic Editor

PLOS ONE

Journal Requirements:

Reviewers' comments:

Reviewer's Responses to Questions

**Comments to the Author**

1. Is the manuscript technically sound, and do the data support the conclusions?

Reviewer #1: Yes

Reviewer #2: Partly

2. Has the statistical analysis been performed appropriately and rigorously? 

Reviewer #1: Yes

Reviewer #2: Yes

3. Have the authors made all data underlying the findings in their manuscript fully available?

Reviewer #1: Yes

Reviewer #2: Yes

4. Is the manuscript presented in an intelligible fashion and written in standard English?

Reviewer #1: Yes

Reviewer #2: Yes

5. Review Comments to the Author

Reviewer #1: In statistical analysis section:

"Following the approach outlined in Kolenikov and Shorrocks

(2005) we take the average of the two effects respectively thus giving a growth effect...."

Why do you take the average of the two impacts, and is there any justification for applying them?

Reviewer #2: 1. In abstract (methods section), author cannot mention what statistical tools has used. It should mention in this section.

2. The author indicated the advantages of this study are that he/she used larger dataset than others studies in Ireland and measured the mental health of same people both before and after Covid. Research gap is not completely clear to me. Please increase the research gap as much as possible.

3. In Table 1, content of the table is vague according to the table title.

4. The authors measured socioeconomic gradient by income and maternal education. Why only choose two variables, income and maternal education? Why not others variables that’s impact on mental health during the Covid pandemic?

5. How can author check the significance of difference between pre-Covid and post-Covid depression rate (Table 3)? What statistical test was performed?

6. How can author compute the transition matrix? Please mention the procedure in the methodology?

7. What are the differences between your study findings and the previous study findings (from literature review)?

8. What are the future implications of this study (since Covid pandemic has already passed)? Please include a recommendation section on the basis of the study findings.

9. Conclusions section is absent in the manuscript. Why?

6. PLOS authors have the option to publish the peer review history of their article (what does this mean?). If published, this will include your full peer review and any attached files.

Reviewer #1: No

Reviewer #2: No

---

## [Author Response · Author response to Decision Letter 0]

10 Jul 2023

Firstly, I would like to thank both reviewers for taking the time to read my paper and for offering very helpful comments. Below I give my response to these comments.

Reviewer 1: 

"Following the approach outlined in Kolenikov and Shorrocks

(2005) we take the average of the two effects respectively thus giving a growth effect...."

Why do you take the average of the two impacts, and is there any justification for applying them?

Response: Thank you. The change in depression rates is decomposed into that part which arises from a change in the average level of depression (as measured by the average of the CESD scores) and from a change in the distribution of the CESD scores, which we label the growth and distribution effects respectively. Such a decomposition requires the choice of a “base” configuration for both of these effects. The choices are (i) to calculate the CESD growth effect with the initial CESD distribution held constant, and to calculate the CESD distribution effect holding the mean CESD at the final level or (ii) to calculate the CESD growth effect with the final CESD distribution held constant, and to calculate the CESD distribution effect holding the mean CESD at the initial level. The choice between approach (i) or (ii) is arbitrary but the precise decompositions will vary. The standard solution to this (as followed in Kolenikov and Shorrocks, 2005, for example) is to take the average of the two effects. I have edited the text to try to make this clearer.

Reviewer 2

In abstract (methods section), author cannot mention what statistical tools has used. It should mention in this section.

Response: I have now included some new sentences which describe the statistical analysis which is carried out.

The author indicated the advantages of this study are that he/she used larger dataset than others studies in Ireland and measured the mental health of same people both before and after Covid. Research gap is not completely clear to me. Please increase the research gap as much as possible.

Response: Thank you. The principal advantage of this study is that we can measure mental for the same people before and during the pandemic. Thus we can identify the change in mental health which occurs during the pandemic. Other studies which monitored mental health during the pandemic, with the exception of Smyth and Murray, provided no information regarding mental health before the pandemic. Thus, using just those studies, there is no way of knowing whether there is a change in mental health following the pandemic, or whether those who suffer from mental health problems in the pandemic had previous conditions. The larger dataset also provides clear advantages in terms of statistical significance and precision of effect sizes. I have inserted extra text to explain this.

In Table 1, content of the table is vague according to the table title.

Response: Thank you, I have edited the table title and I think it is clearer now.

The authors measured socioeconomic gradient by income and maternal education. Why only choose two variables, income and maternal education? Why not others variables that’s impact on mental health during the Covid pandemic?

Response: The analysis of how any health condition varies by socioeconomic status (SES) is a standard exercise in health economics and epidemiology. To carry out such analysis we need a measure of SES which is measured accurately and which is clearly ranked (ranking is necessary to calculate concentration indices), in the sense that given any two observations it is always possible to rank one as having a higher SES than the other, or that they are equal. While there may be other variables which influence mental health during the pandemic, if they are not clearly a measure of SES and cannot be ranked (e.g. social class is a measure of SES but it can be difficult to provide a clear ranking). The two variables available in the Growing Up in Ireland survey (GUI) which are best suited for measuring SES are maternal education and after-tax income. I have elaborated further on this in the text.

How can author check the significance of difference between pre-Covid and post-Covid depression rate (Table 3)? What statistical test was performed?

Response: This analysis was carried out using the DASP programme in Stata (DASP: Distributional Analysis Stata Package, Abdelkrim Araar, Jean-Yves Duclos, Université Laval, PEP and World Bank, December 2021). The statistical test is a standard difference in proportions test, which I now mention in the text.

How can author compute the transition matrix? Please mention the procedure in the methodology?

Response: The transition matrix is an empirical transition matrix and the graphical version of it is calculated directly from the data using the tabplot command in Stata. The underlying data used to construct it can be obtained from a cross tabulation of the data by CESD score and GUI wave. The matrix is a nxn matrix, the column shows the fraction of the sample with that CESD score pre-Covid, and the row indicates the fraction of the sample with that CESD score post-Covid. The entries along the main diagonal of the matrix indicate the fraction whose CESD score does not change pre and post Covid. I have edited the methodology section to explain this.

What are the differences between your study findings and the previous study findings (from literature review)?

Response: Thank you. Our study differs from other Irish studies mentioned in the literature review in explicitly taking account of the dynamics of depression by looking at the change in CESD scores pre and post Covid. It also explicitly examined the socioeconomic gradient associated with CES-D8 scores pre and post pandemic. The study also differs in that it decomposes the overall change in depression into growth and distribution effects These issues are covered in the discussion and I have expanded on this in the text.

What are the future implications of this study (since Covid pandemic has already passed)? Please include a recommendation section on the basis of the study findings.

Response: Thank you. The study is primarily descriptive and I am wary of drawing strong policy conclusions from it, especially as the circumstances of Covid-19 may be unique and even if we have future pandemics (which seems likely) their precise nature may differ from Covid 19. I have included a new paragraph at the end of the “Discussion” section which covers these issues.

Conclusions section is absent in the manuscript. Why?

Response: Thank you. The conclusion section was effectively the last paragraph of the “Discussion” section. I have now re-organised this so that there is a separate, short, conclusions section.

---

## [Decision Letter · Decision Letter 1]

18 Oct 2023

PONE-D-22-33706R1Mental Health in Ireland During the Covid Pandemic: Evidence from Two Longitudinal SurveysPLOS ONE

Dear Dr. Madden,

Thank you for submitting your manuscript to PLOS ONE. After careful consideration, we feel that it has merit but does not fully meet PLOS ONE’s publication criteria as it currently stands. Therefore, we invite you to submit a revised version of the manuscript that addresses the points raised during the review process.

The manuscript has been evaluated by two reviewers, and their comments are available below.

The reviewers have raised a number of major concerns. They feel the manuscript should outline a clearly-defined research question, and they request improvements to the reporting of methodological aspects of the study, for example, regarding the exclusion criteria and more information on how the data collection was completed. The reviewers also note concerns about the statistical analyses presented and request re-analyses be completed.

Could you please carefully revise the manuscript to address all comments raised?

We look forward to receiving your revised manuscript.

Kind regards,

Avanti Dey, PhD

Staff Editor

PLOS ONE

Reviewers' comments:

Reviewer's Responses to Questions

**Comments to the Author**

1. If the authors have adequately addressed your comments raised in a previous round of review and you feel that this manuscript is now acceptable for publication, you may indicate that here to bypass the “Comments to the Author” section, enter your conflict of interest statement in the “Confidential to Editor” section, and submit your "Accept" recommendation.

Reviewer #2: All comments have been addressed

Reviewer #3: (No Response)

2. Is the manuscript technically sound, and do the data support the conclusions?

Reviewer #2: Yes

Reviewer #3: No

3. Has the statistical analysis been performed appropriately and rigorously? 

Reviewer #2: Yes

Reviewer #3: No

4. Have the authors made all data underlying the findings in their manuscript fully available?

Reviewer #2: Yes

Reviewer #3: Yes

5. Is the manuscript presented in an intelligible fashion and written in standard English?

Reviewer #2: Yes

Reviewer #3: No

6. Review Comments to the Author

Reviewer #2: (No Response)

Reviewer #3: Abstract

The sentence beginning with…’For mothers from the 2008 cohort..’ may be recast to read better e.g..For mothers from the 2008 cohort, a gradient was observed during the pre-COVID-19 pandemic period with poorer mental health status for lower-income and less educated mothers.

The conclusion must be more comprehensive than just two lines. What are the implications of findings from this study?

Introduction:

Paragraph 3, PCs should be put in full at the first mention in the introduction. Definition appearing in the abstract doesn’t count here.

The discussion of GUI data in the introduction is too much. Most of the aspects of the GUI data should be moved to the methods section. A mention in one or two lines of the data at the end of the introduction section may be appropriate.

The presentation of literature can be improved. The author may not report extant studies one by one. Synthesis of information of information in a more concise and comprehensive manner is recommended.

The author can either be passive or use the first person in the presentation since they are single author. The use of ‘we’ and ‘our’ as in ‘our paper’ doesn’t make sense.

Methods:

Research design: this study cannot be cross-sectional because the data collection points are more than one over time. Cross-sectional designs entail a snapshot i.e gathering information at a single point in time.

Mentioned that the CES-D scale has been used in several publications previously. It would be good to cite one or two of these.

Results:

Table titles are to be put on top of the table not under it. Additionally, it must include the total number of respondents (N = ?)

Table 1 is confusing. The first part is relating maternal education and gender. Doesn’t maternal education entail only women?

Figures 10 and 11 mentioned are not appearing in the text.

How does the methods used control for the effects of other variables to the mental health status?

May need to work on the language used to reflect the professionalism required in publications of this nature. Words such as ‘alas’ ‘disturbing’ could be edited out.

Discussion:

In paragraph 1, the contribution of the study cannot be availability of the data. The data is already available whether or not the study was conducted!

Discussion reports the findings more than it discusses the implications of the findings. Also no proper engagement with the literature in the field.

7. PLOS authors have the option to publish the peer review history of their article (what does this mean?). If published, this will include your full peer review and any attached files.

Reviewer #2: No

Reviewer #3: No

---

## [Author Response · Author response to Decision Letter 1]

24 Oct 2023

Below I give my response to the specific comments raised by referee 3 and where I have not fully complied with the request, I have indicated why not. 

I include the comment in italics and my response in non-italic font.

The sentence beginning with…’For mothers from the 2008 cohort..’ may be recast to read better e.g..For mothers from the 2008 cohort, a gradient was observed during the pre-COVID-19 pandemic period with poorer mental health status for lower-income and less educated mothers.

The conclusion must be more comprehensive than just two lines. What are the implications of findings from this study?

Response: Thank you, I have incorporated these edits and extended the “conclusion” section.

Paragraph 3, PCs should be put in full at the first mention in the introduction. Definition appearing in the abstract doesn’t count here.

Response: Thank you, this edit has been carried out.

The discussion of GUI data in the introduction is too much. Most of the aspects of the GUI data should be moved to the methods section. A mention in one or two lines of the data at the end of the introduction section may be appropriate.

Response: Thank you. I have removed most of the discussion of the data to the Methods section.

Research design: this study cannot be cross-sectional because the data collection points are more than one over time. Cross-sectional designs entail a snapshot i.e gathering information at a single point in time.

Response: Thank you, I replaced the words “cross-sectional” with “longitudinal”.

Mentioned that the CES-D scale has been used in several publications previously. It would be good to cite one or two of these.

Response: Thank you, I have included two extra references to cover this.

The presentation of literature can be improved. The author may not report extant studies one by one. Synthesis of information of information in a more concise and comprehensive manner is recommended.

Response: Thank you, I have re-drafted this section along the lines suggested. The review now covers evidence for the mental health effects of the initial stages of the pandemic, for the later stages of the pandemic, and also some studies for Ireland.

The author can either be passive or use the first person in the presentation since they are single author. The use of ‘we’ and ‘our’ as in ‘our paper’ doesn’t make sense.

Response: Thank you. I think this could be viewed as mostly a matter of style. The use of the plural pronoun (we) rather than singular (I), even with a single author, while unusual, is not unprecedented. However, having no strong views on the matter, I have changed to passive mode. 

Table titles are to be put on top of the table not under it. Additionally, it must include the total number of respondents (N = ?)

Response: Thank you, this has been done. I have interpreted the comment regarding sample size as applying to table 1. I have not replicated this information for other tables.

Table 1 is confusing. The first part is relating maternal education and gender. Doesn’t maternal education entail only women?

Response: Thank you. I have now changed this to “Principal Carer’s education” and have done this throughout the manuscript. 

Figures 10 and 11 mentioned are not appearing in the text.

Response: I am sorry, I think this comment may be mistaken. Figures 10 and 11 appear in the file I submitted “Mental Health in Ireland – figures only.

How does the methods used control for the effects of other variables to the mental health status?

Response: The methods used here are not regression methods and hence controls for a range of covariates are not used. The analysis is stratified by gender and principal carer’s education and the calculation of the concentration indices also shows how the dependent variable varies across income. In addition, as this is longitudinal data, the analysis is automatically controlled for time invariant variables.

May need to work on the language used to reflect the professionalism required in publications of this nature. Words such as ‘alas’ ‘disturbing’ could be edited out.

Response: Thank you. Again, perhaps a matter of style, but I have made the edits suggested.

In paragraph 1, the contribution of the study cannot be availability of the data. The data is already available whether or not the study was conducted!

Response: Thank you, you are quite right! I have edited this to saying that it is the use of these measures which is the contribution. To the best of my knowledge no other study has exploited this feature of GUI data.

Discussion reports the findings more than it discusses the implications of the findings. Also no proper engagement with the literature in the field.

Response: I think it is fair to say that most discussions usually start off with a brief recap of the main results, but then adding context and discussion. The decision to be conservative in terms of reference to previous studies in the literature arises because of the belief that Covid 19 was an event with little or no precedent. However, I have made some edits which have put the results of this study in the context of other literature in the area. Thus there is now a greater stress on the unusual finding that there was little if no socioeconomic gradient associated with the deterioration in mental health. I have also discussed my results in the context of other results regarding differences in mental health by gender e.g. the studies by Vloo et al and also Riecher-Rössler.

In terms of the implications of the findings, I think I have discussed these but I have been reluctant to stress them too much as I believe it is still unclear as to how generalizable results from the Covid pandemic will be. One of the lessons of Covid was that the experience from some previous public health epidemics such as influenza were of limited help. One of the main contributions of this paper is to show, in terms of mental health at least, what sort of data should be collected on a regular basis and which could be of use for future pandemics, and I have included an extra sentence stressing this.

---

## [Decision Letter · Decision Letter 2]

16 Nov 2023

Mental Health in Ireland During the Covid Pandemic: Evidence from Two Longitudinal Surveys

PONE-D-22-33706R2

Dear Dr. Madden,

We’re pleased to inform you that your manuscript has been judged scientifically suitable for publication and will be formally accepted for publication once it meets all outstanding technical requirements.

Kind regards,

Tracey Smythe, PhD

Academic Editor

PLOS ONE

Reviewers' comments:

Reviewer's Responses to Questions

**Comments to the Author**

1. If the authors have adequately addressed your comments raised in a previous round of review and you feel that this manuscript is now acceptable for publication, you may indicate that here to bypass the “Comments to the Author” section, enter your conflict of interest statement in the “Confidential to Editor” section, and submit your "Accept" recommendation.

Reviewer #2: All comments have been addressed

Reviewer #4: All comments have been addressed

2. Is the manuscript technically sound, and do the data support the conclusions?

Reviewer #2: Yes

Reviewer #4: Yes

3. Has the statistical analysis been performed appropriately and rigorously? 

Reviewer #2: Yes

Reviewer #4: Yes

4. Have the authors made all data underlying the findings in their manuscript fully available?

Reviewer #2: Yes

Reviewer #4: Yes

5. Is the manuscript presented in an intelligible fashion and written in standard English?

Reviewer #2: Yes

Reviewer #4: Yes

6. Review Comments to the Author

Reviewer #2: (No Response)

Reviewer #4: This is a cogent well-written paper highlighted the potential greater vulnerability of women in Ireland to a decline in their mental health in the context of the COVID-19 pandemic.

7. PLOS authors have the option to publish the peer review history of their article (what does this mean?). If published, this will include your full peer review and any attached files.

Reviewer #2: No

Reviewer #4: **Yes: **Dr Sarah Markham

---

## [Editor Report · Acceptance letter]

24 Nov 2023

PONE-D-22-33706R2 

Mental Health in Ireland During the Covid Pandemic: Evidence from Two Longitudinal Surveys 

Dear Dr. Madden:

I'm pleased to inform you that your manuscript has been deemed suitable for publication in PLOS ONE. Congratulations! Your manuscript is now with our production department. 

Kind regards, 

on behalf of

Dr. Tracey Smythe 

Academic Editor

PLOS ONE